# Providing Access to Urban Green Spaces: A Participatory Benefit-Cost Analysis in Spain

**DOI:** 10.3390/ijerph17082818

**Published:** 2020-04-19

**Authors:** Silvestre García de Jalón, Aline Chiabai, Alyvia Mc Tague, Naiara Artaza, Amaia de Ayala, Sonia Quiroga, Hanneke Kruize, Cristina Suárez, Ruth Bell, Timothy Taylor

**Affiliations:** 1Basque Centre for Climate Change (BC3), 48940 Leioa, Spain; aline.chiabai@bc3research.org (A.C.); alyvia.mctague@ucdconnect.ie (A.M.T.); naiara.artaza@ehu.eus (N.A.); amaia.deayala@bc3research.org (A.d.A.); 2Department of Applied Economics I, University of the Basque Country (UPV/EHU), 48940 Leioa, Spain; 3Department of Economics, Universidad de Alcalá, 28801 Alcalá de Henares, Spain; sonia.quiroga@uah.es (S.Q.); cristina.suarez@uah.es (C.S.); 4National Institute for Public Health and the Environment (RIVM), P.O. Box 1, 3720 BA Bilthoven, The Netherlands; Hanneke.kruize@rivm.nl; 5Department of Epidemiology & Public Health, University College London, London WC1E 7HB, UK; r.bell@ucl.ac.uk; 6European Centre for Environment and Human Health, University of Exeter Medical School, Truro Campus, RCH Treliske, Truro TR1 3HD, UK; timothy.j.taylor@exeter.ac.uk

**Keywords:** green space, participatory benefit-cost analysis, participatory evaluation, cost-benefit analysis

## Abstract

The opening up of green spaces could provide significant benefits to society. This study develops a framework to assess the economic benefits and costs of public interventions providing citizen access to urban green spaces. The Thinking Fadura project in Getxo (Spain) was used as a case study. A method for participatory benefit-cost analysis is developed, where a stakeholder-participatory evaluation is combined with a standard cost-benefit analysis. The participatory evaluation followed a bottom-up approach in a sequential evaluation including three main focal points: key stakeholders and experts, visitors and the general public. The assessment demonstrates that the Thinking Fadura project’s benefits outweigh the costs. The results suggest that projects designed with the purpose of improving green space accessibility to the general public can be beneficial from a societal perspective. The highest economic benefits were an increase in the amenity and recreational value and an increase in people’s physical activity. The participatory evaluation indicates that giving access to people of lower socio-economic status and vulnerable groups and improving recreational use were perceived as the most beneficial. An increase in noise, dirt, and risk of criminal activities as well as potential conflicts between green space users were perceived as the most negative impacts of opening a previously restricted area to the general public. The economic assessment of Thinking Fadura project could serve as a model in the decision-making process in locations where the use of greenspaces is restricted.

## 1. Introduction

The proportion of people living in urban areas is expected to rise from 46.6% in 2000 to 69.6% in 2050 [1]. As a result of this increase in urbanisation, green space in urban areas is facing increasing pressure. In this context, there are significant challenges faced by planners and local authorities in the balancing of the demands for space and development with the needs of the population for green spaces.

To counteract the negative effects of increased urbanization, movements such as “Healthy Cities” started by the World Health Organization have taken root in many urban areas. This movement is a long-term international development initiative that promotes physical and social environments by putting health high on decision makers’ priorities [2]. Part of improving accessibility to healthy environments is to promote healthy living, which includes physical and mental well-being, as well as increased access to spaces that promote physical activity. Green spaces are connected to psychological and physical benefits through their use as spaces for physical activity and mental restoration [3,4,5].

Urban green spaces can provide multiple benefits to human health by fostering an increase in physical activity [6,7], improving air quality [8], reducing noise pollution [9], and protecting against high temperature and heavy rainfall [10]. Apart from physical health, urban green spaces contribute to people’s wellbeing and mental health by increasing recreation activities [11], reducing stress [5] and stimulating social contacts and increasing societal cohesion [12].

In terms of climate change mitigation and adaptation, urban green spaces play an important role first by sequestering carbon and reducing atmospheric CO_2_ concentrations and then by lowering the higher temperatures that may be accentuated by the “urban heat island” effect. This phenomenon is when urban zones tend to be hotter on warm days than their surrounding areas [13]. One of the reasons this may result in urban areas is the lack of vegetation to shade and cool the air.

The INHERIT project (www.inherit.eu) investigated whether investing in making urban green spaces accessible to all has potential to contribute to a triple win of improving health, improving equity and contributing to environmental sustainability. Bell et al. [6] described important lessons for good practice in changing contexts to modify behaviours and achieve this triple win. Aligned with Kruize et al. [12], they highlighted the contribution of green spaces to well-being, and the fact that creating more readily accessible and good quality green space often provides opportunities to socialise and to be active, as well as contributing to mitigate climate change impacts. Apart from having a beneficial mental health effect, green spaces can moderate the association of psychological distress with area-level socio-economic status. Sugiyama et al. [5] found that the odds of higher psychological distress was higher in residents in lower socio-economic status areas than those in higher socio-economic status areas.

Economic analyses can be used to demonstrate quantitatively the costs and benefits of green spaces to policy makers and society. Municipal budgets for green space management are often under pressure and consequently there is an urgent need for new funding models to demonstrate the need for urban green spaces. Inspired by the TEEB study [14], van Zoest and Hopman [15] described an initiative to develop a tool that makes the financial benefits of green spaces visible in the municipal balance sheet. Harnik and Welle [16] set forth a methodology for valuing the benefits of urban parks. They found seven major factors that could be quantified and valued in parks: property value, tourism, direct use, health, community cohesion, clean water, and clean air. Researchers [17,18] estimated the non-market benefits derived from the provision of a new urban park where there was an old train station. Through a contingent valuation method, they found that the mean willingness to pay was considerably higher for people who live closer to the planned park as it was more accessible to them. Cheng and Wang [19] showed the economic benefits of green spaces by empirically assessing the evolving path of urban green spaces along with economic development and urbanization through panel data across 285 Chinese cities. The authors found an N-shaped environmental Kuznets curve, indicating that green space coverage increased at the initial stage of economic development, and then it started to decrease as GDP per capita exceeded a certain threshold and then increased again at a high GDP per capita level.

Economic evaluations through cost-benefit analyses (CBA) typically estimate the total value of a particular green space (e.g., [16,17,18]) and do not focus on their specific attributes such as the value of improved accessibility or extended exposure to green spaces. As far as the authors are aware there are no studies in the scientific literature that economically assess the benefits of providing or improving green space access that previously was restricted to the general public. This paper addresses this gap in the literature, by evaluating the economic benefits and costs of a project called Thinking Fadura aimed at increasing access to restricted green spaces. The other main literature gap that is addressed in this study is the direct participation of key stakeholders, users of the green spaces and the general public in the processes for designing and implementing the CBA. Previous studies have tried to incorporate stakeholders’ views in similar methodologies such as Multi-Criteria Decision Making [20]. Our assessment incorporates a participatory process into a CBA and shows the synergies of combining them for future economic evaluations. Similar approaches based on the participation of stakeholders have been used to allow communities to compare the costs and benefits of public interventions and/or identify disaster risk reduction measures (e.g., [21]). Campos et al. [22] found that a multidirectional engagement process with diverse stakeholder groups and at various scales (stakeholders, users and general public) ensures future social and sustainable acceptability. Thus the combination of participatory process and a CBA enhances the robustness of the results and their social acceptability.

### Background

The Thinking Fadura project in Getxo, Basque Country (Spain) was used as a case study. The project aimed to design a new public space where sporting facilities coexisted with open green spaces. The main idea was a combination of a nature-based solution with sporting facilities to foster physical activity along with access to green spaces.

The initiative was the result of a community reflection based on a participatory design methodology carried out in 2017 where participants proposed to open the greenspaces of Fadura’s Municipal Sports Center (FMSC) to the general public. In the past, only people who were registered and paid the annual fee had access to the area. FMSC’s facilities occupy around 20 hectares along the Gobela River. It has many green spaces and numerous sports facilities such as indoor and outdoor swimming pools, gym, soccer fields, rugby, basketball, tennis and paddle tennis, among others. The opening of the park was conceived as an urban and landscaping project. The new public park had 8.2 hectares of green land surrounded by both public and private sports facilities. At the time when the assessment was performed in 2018, only around five hectares of the greenspaces were open to the general public. The remaining green spaces were closed due to the ongoing construction works which were expected to take three years more. After the project implementation, users willing to access private facilities had the opportunity to do so by buying a one-day ticket.

Getxo is a coastal town and had 77,946 inhabitants in 2019. Getxo is a mostly affluent residential area, as well as being the third largest municipality of Biscay province. In regards to the ecological characteristics of the area, Fadura is located in a very important strategic position for Getxo. The park’s area, works like a hinge between the dense urban district, the coast, the Gobela river, and the natural environment surrounding it (the Bolue wetland). It is essential to highlight the role that the river Gobela and the Bolue wetland have in Getxo, both strictly related with Fadura, as biodiversity reserves and possible axes for a greener city. The geographical location of the sports centre of Fadura has a strong impact on its character, not only at the environmental and natural level but also, at the level of human geography, urban planning and accessibility. The redesign of the park is a unique opportunity to diversify the activities that can be carried out in this natural environment, maintaining the continuity of the ecosystem along the ecological corridor that forms the river and trying to reach a sound balance between accessibility for leisure and conservation of the ecosystem, respecting the native fauna and flora.

The main objective of the project is to increase the accessibility and usage of green spaces, as a way to promote physical activity, to facilitate social cohesion, and to bring citizens closer to the natural environment. The implementation of the project not only allows the general public to get access and enjoy the green spaces surrounding FMSC, but also serves to cross and connect the city, like a green belt. In terms of public works, opening the Fadura park implied planting trees along the path, creating paths that crossed the park and connected it with existing walking paths and biking tracks, promoting public access to certain multipurpose sports facilities and creating leisure facilities as a way to promote social interaction and cohesion.

## 2. Methods

### 2.1. Participatory Benefit-Cost Analysis (PBCA) and Data Collection Methods

Cost–benefit analysis (CBA) is a recognized analytical tool of economic analysis for decision-making and offers a method for comparing in a quantitative way the benefits against the costs of a particular investment [23,24,25,26,27]. There are two main types of CBA, financial and economic. Whilst the financial CBA only includes market costs and benefits, the economic CBA includes both market and non-market costs and benefits from a societal perspective. The latter is the case of this study.

CBA can suffer from various limitations such as potential inaccuracies and subjective judgements involved when identifying, quantifying, and estimating costs and benefits [28]. Another limitation is the complexity associated with estimating some intangible benefits such as equity or societal cohesion, or other community-based values, for which the translation in monetary terms is problematic or would pose potentially controversial assumptions. A way to address these limitations is to incorporate key actors, stakeholders and the general public in the co-design process of the CBA and complement the economic assessment with additional qualitative indicators based on citizens’ perceptions. This combination of both a participatory evaluation and a CBA is called Participatory Benefit Cost Analysis (PBCA). Campos et al. [22] defined the PBCA as *“a hybrid methodology of economic project appraisal as it is composed of heterogeneous sources and diverse elements, combining interpersonal deliberation and quantitative methodologies to produce both depth and breadth in valuation and appraisal processes”*.

PBCA includes the traditional CBA and, additionally, seeks to achieve a high level of consensus about the identification, categorization and estimation of the costs and benefits among the participants involved in decision-making processes. In the PBCA, the participatory evaluation is highly important for the robustness and quality of the evaluations, and for increasing social acceptability to support implementation. There exists a wide range of participatory approaches that can be included in the PBCA such as citizen surveys; stakeholder analysis, interviews and workshops; focus groups; systemization of experiences; fuzzy cognitive mapping and participatory state of the art [22].

Similar to the traditional economic CBA, benefits and costs in the PBCA are calculated in monetary terms and are adjusted for the time value of money. In order to account for future generations and their interests, the discount rate for societal costs and benefits *j* is adjusted from the private market rate [29] as the latter could underestimate the social benefits of long-term interventions. The social discount rate is usually lower than the private one, which supports benefits and investments for future generations. The Net Present Value (NPV) is used to express all flows of benefits and costs over time on a common basis by taking into consideration the time when they are incurred (Equation (1)):(1)$$NPV=\overset{T}{\underset{t=0}{\sum }}\left(\frac{\left(B_{t}-C_{t}\right)}{{\left(1+r\right)}^{t}}\right)+\overset{T}{\underset{t=0}{\sum }}\left(\frac{BE_{t}-CE_{t}}{{\left(1+j\right)}^{t}}\right)$$
where *BE_t_* represents the value of the non-marketed benefits in monetary terms and *CE_t_* the values of the non-marketed costs in monetary terms.

Other main indicators for measuring the economic feasibility in CBAs are the Internal Rate of Return (IRR), the Benefit-Cost Ratio (BCR) and the Payback Period (PBP) [30]. This study has conducted a PBCA to evaluate the economic costs and benefits of implementing the Thinking Fadura project (opening the green spaces to the general public) from the point of view of the society as a whole. Figure 1 shows the conceptual framework to evaluate the Thinking Fadura project. Within the PBCA, the participatory evaluation was linked to the CBA with the purpose of providing information that could not have been included in a traditional CBA. Ad hoc stakeholders’ participation process and citizens’ surveys were put in place to gather specific information needed for the evaluation of quantitative aspects, such as categories of benefits and risks, current and use, perceptions on impacts, attitudes, or socio-economic factors.

### 2.2. Participatory Evaluation and Data Collection Methods

The participatory evaluation followed a bottom-up approach in a sequential evaluation including three main focal points of the analysis: key stakeholders and experts, visitors and the general public. This assessment may provide a better understanding of the needs of implementing Thinking Fadura and their impacts. The participation of stakeholders and experts, visitors and the general public also helped to design a more precise economic evaluation.

#### 2.2.1. Key Stakeholders and Experts

The first step in the PBCA was the assessment with key stakeholders and experts. A workshop with 20 participants was held in May 2018, including members from public services of different areas such as social welfare, equality, multiculturalism, development cooperation, environment, urban planning, housing, civil protection, economic promotion and health and Fadura sporting area. In the development of the workshop there were two people facilitating the session.

The workshop had four objectives: (i) to provide a better understanding of the context of Thinking Fadura, (ii) to identify the elements and the population affected by opening the green spaces, (iii) to identify new uses in the green spaces and relate them with potential impacts and (iv) to assess the potential impacts as a result of increasing accessibility in the green spaces (more details in the description of the workshop in the Appendix A).

#### 2.2.2. Visitors

Subsequently, an assessment focusing on users of the green spaces of Fadura was carried out between August 2018 and June 2019. This included counting the users, an in-situ survey to visitors and an assessment of physical activity.

##### Counting the Users

The counting of visitors was done through manual counters at different times of the day, days of the week and months of the year. The counting of visitors was needed for quantifying the recreation benefit by the travel cost method [31].

##### In-Situ Survey to Visitors

The in-situ survey to visitors provided quantitative data on frequency of use and potential increase in the use of the park, travel cost values, and perceptions about the benefits and costs of Thinking Fadura (for more detail see the Appendix A). Through this survey we obtained both qualitative and quantitative information that was used for the different analysis carried out in this study. The target population was adults who transited through the green areas that were currently open to the public. A total of 256 visitors over 16 years old of age were surveyed.

##### Physical Activity Assessment

The assessment of physical activity was done with the System for Observing Play and Active Recreation in Communities (iSOPARC, [32]) tool (see the SOPARC assessment in the Appendix A). iSOPARC is an established and validated observation tool to assess the use of public spaces in community settings [32]. iSOPARC is used to record individual (gender, age, physical activity level, and ethnicity) and contextual characteristics (in a given area) and primary activity for each observed person. Through visual scans, iSOPARC determines the usage and physical activity of the greenspace at a determined time.

In Thinking Fadura, the iSOPARC assessment was carried out in order to determine the number of people (all ages) increasing physical exercise as a result of the restoration of the pathway and the improved accessibility in Fadura.

The green areas of Fadura (around 4 ha) were divided in five target areas (see Appendix A). The observations were done by two observers in August 2018. In total sixteen site visits were done in each Target Area (5 TAs) at different time of the day. These times were at 9.00 am, 12.00 pm, 5.00 pm and 8.00 pm. The site visits were done in twelve different days covering both weekdays and weekends. Appendix A shows the temporal distribution at different times. Every site visit covered the five target areas.

In the five target areas, the iSOPARC [32] tool counted an average of 2.05 people per scan. For the physical exercise benefit only those people who were doing walking or vigorous activities were considered. Thus in the five target areas, the average of people increasing physical exercise for walking and vigorous activities were 1.06 and 0.21, respectively (Appendix A). Hence, the sum of people increasing physical exercise was 1.27 (people/iSOPARC scan).

#### 2.2.3. General Public

The final step in the PBCA was the assessment of opinions among the general public. This was done through a face-to-face survey in the municipality of Getxo (see the description of the survey to the general public in the Appendix A). The aim of this survey was to study the opinion of the citizens of Getxo with regards to Thinking Fadura´s initiative of opening Fadura´s green spaces to the general public. The target population was people over 16 years old and registered as citizens of the municipality of Getxo. To identify this population, two screening questions were included in the introduction of the survey. In total, 250 surveys were conducted in 12 different areas of the municipality.

### 2.3. Cost-Benefit Analysis (CBA): Methods for Assessing Impacts

This section describes the calculation of the benefits and costs in the PBCA which were selected through the participatory evaluation and literature review.

#### 2.3.1. Benefits

##### Reduction in Pollutant Emissions Associated with Reduction in Car Use

Before the project was implemented the green spaces of Fadura were fenced and people could only enter through the main gate of the sporting facilities. After the project implementation, people could then cross the area through the new pathway. Hence, it was expected that people who live in the surroundings could sometimes walk through this new pathway instead of driving for short distances. This produces a reduction in pollutant emissions due to people using fewer cars as a result of the built pathway in Fadura (Equation (2)). In the CBA, two categories of pollutant emissions were considered. The first group was related to a reduction of CO_2_eq emissions and the second group was related to toxic air pollutants (NO_x_, PM and SO_2_).
(2)$$VP_{ij}=\Delta km_{j}\times E_{i}\times EV_{i}$$
where $VP_{ij}$ is the value of reduction in pollutant *i* and year *j* ($\frac{€}{\mathrm{year}}$), $\Delta km_{j}$ is the total reduced km in year *j* ($\frac{\mathrm{km}}{\mathrm{year}}$), $E_{i}$ is the emission factor per km for pollutant *i* ($\frac{\mathrm{kg}}{\mathrm{km}}$), and $EV_{i}$ is the externality value per kg of pollutant *i* ($\frac{€}{\mathrm{kg}}$).

The total number of reduced vehicle kilometres was estimated as the product of the reduced number of cars as a consequence of people walking instead of driving, and the average distance (Equation (3)):(3)$$\mathrm{RC}\;=\;\mathrm{PWD}\;\times \frac{P}{PC}$$
where RC is the reduced number of cars per year, *P* is the percentage of people walking instead of driving as a result of Thinking Fadura and *PC* is the assumed number of people per car.

The number of new visitors in Fadura and the percentage of visitors who would walk instead of driving was obtained from the survey to visitors. Based on the knowledge of the area and interviews with visitors, the average reduced distance including return trip was assumed to be 5 km per car. It was considered that when people needed to do long distances they would use the car instead of walking. The amount of pollutant and CO_2_eq emissions per kilometre was estimated on the basis of the average emissions level of a new car sold in 2017. An approximate average emissions level per kilometre is 118.5 g of CO_2_, 0.4 g of NO_x_, 0.00629 g of SO_x_ and 0.02857 g of PM [33].

Following the UK Department for Energy and Climate Change [34], the externality value of reducing one ton of CO_2_eq was €14.75. Although DECC predicted that these values would increase over time up to €93 (t CO_2_eq)^−1^ in 2030, the current value was used in this study. In the case of the other pollutants, the valuation was based on the damage cost approach (dose-response method) which focuses on the quantification of the explicit impact that the emissions have on human health, environment and economic activity [35]. In Spain the externality values were €4964 (t NO_x_)^−1^, €7052 (t SO_2_)^−1^ and €48,012 (t PM)^−1^.

##### Noise Pollution Reduction as a Result of Lower Use of Cars

Noise emissions from traffic can produce health and environmental problems [35]. In this study the benefit refers to the health effect due to people using fewer cars as a result of the built pathway in Fadura. Reduction in noise pollution was calculated as the product of the total km reduced and the noise pollution reduction externality value. The valuation of the noise externality value per km was based on the bottom-up estimates of marginal noise costs from [36]. These values depend on the type of vehicle, type of traffic and population density exposed to the noise. The noise externality value was €0.001325 per km and was calculated as the average of values in suburban areas during daylight and night times.

##### Fuel Savings as a Result of People Using less Cars

Fuel savings refers to the economic benefit associated with a lower use of cars as a result of the built pathway in Fadura. As cars will be used less, less money will be spent on fuel and people will save money for other purposes (Equation (4)):(4)FS =Δkm× PropC × DC × FPr
where FS indicates the fuel savings measured in euros per year, Δkm is the reduced vehicle kilometres, PropC the proportion of cars (%), DC the diesel consumption per km and FPr the fuel price. *f* indicates whether the car was diesel or gasoline. It was considered that the proportion of diesel and gasoline cars was 67% and 33% in Getxo ([37], according to the EU car sales). The diesel price assumed was 1.197 (€/L diesel). Mean diesel consumption per km (L diesel/km) in Getxo was estimated to be 0.07 L/km, which was calculated as the sum of official diesel consumption and the difference between official and real diesel consumption. The gasoline price assumed was 1.236 (€/L gasoline). Mean gasoline consumption per km (gasoline L/km) was estimated to be 0.08 (Ll/km).

##### Amenities and Recreation

To value the amenities and recreation benefits we used two different methodologies according to the distance of respondents’ residence from the green park. For those who live close to the park (≤200 m) we used the hedonic pricing method, drawing on property values, while for those further away (>200 m) we used the travel cost method. We use these approaches separately to avoid double counting.

Increased property value method (≤200 m).

Previous research showed that people are willing to pay more for houses that are close to green spaces and natural areas [16,38,39]. The hedonic pricing method relates individual utility to a set of characteristics of the house [40], including physical characteristics of the building, public transport and other public facilities, neighbourhood features (distance to shops, city centre, etc.), environmental quality and amenity values (parks, trees, water bodies, recreational areas). In this study, we used the increased property value as a proxy for the hedonic price. The increase in property value as a result of an improved access to green spaces was calculated following Equation (5) to capture recreation and amenity values for people living in less than 200 m:(5)IPV =HA× PrIH × PrH
where IPV is the increased property value, *HA* is the total number of houses positively affected by Thinking Fadura, PrIH is the price increase of houses (%) and PrH is the average price of houses.

According to Bianchini and Hewage [39], property values in urban areas with green spaces nearby could have a general increase of 15–25%. Luttik [41] concluded that with a view of open space, the house price would increase by 6–12%. In the literature no study was found with an estimation of a premium on accessible vs. non-accessible green space (private parks not accessible to the general public). Thus, in Thinking Fadura, only an increase in property value of 1% percent was considered. This is explained due to fact that these valuations focused on the benefit of green spaces, whereas in the valuation of Thinking Fadura, only improved accessibility to green spaces is measured.

Some authors have found that the increase in property value decreases with distance away from the green spaces. Harnik and Welle [16] suggested that this effect can be measured up to around 700 m from the green space, with the greatest value found within first 150 m. In our study, it was considered that only those houses in a distance shorter than 200 m would be benefited from the improved access to the green spaces of Fadura. Within the 200-m distance, a total of 600 houses were counted. The average price of these houses was calculated from a state agency. 103 houses were found to be on sale in a distance shorter than 200 m. The average price of all houses according to the local state agencies (www.idealista.com and www.fotocasa.es) was €344,134 per house.

Travel cost method (≥200 m).

The valuation of the recreational use of the park was complemented by the travel cost method for all respondents living at a distance higher than 200 m from the park. This is to avoid double counting, as the increased property value calculated in the previous section already included the recreational value for houses in the radius of 200 m around Thinking Fadura. In fact, hedonic prices measure the consumer surplus from the recreational use and the corresponding amenity value from living near an urban park, while the travel cost method measures only recreation and not the amenity value [42].

The recreation value of a natural ecosystems is usually associated with the direct use that individuals make of these natural assets [43]. The recreational value was estimated by using the travel cost method. Using this method, it was possible to estimate the demand curve for each travel method (on foot, by bicycle, by car and by public transport) and to calculate the consumer surplus value. This method assumes that the costs and time that people incur during a recreational trip to a ‘natural resource’ site can be used to infer the recreation value of that site (Equation (6)):(6)$$R\;=\overset{4}{\underset{i=1}{\sum }}P\;\times {\;\mathrm{TC}}_{i}\times {\;\mathrm{PropV}}_{i}$$
where R indicates the recreation value in euros per year, P is the number of people per year, TC indicates the travel cost in euros per person and PropV the proportion of visitors. The sub-index *i* indicates the travel methods (on foot, by bicycle, by car and by public transport). The mean number of users for recreation was estimated from the counting done during the face-to-face survey implementation to visitors. Again, the time during the year during rain events was excluded from the analysis.

The travel cost was calculated for the different types of visitors according to the transport means used to get to Fadura. From the survey to visitors, the proportion of visitors on foot, by bicycle, by car and by public transport was estimated (Table 1). A time opportunity cost of €10 per hour was considered [44]. In order to avoid double counting, visitors who lived within the area considered for the estimation of the increased property value were excluded in the estimation of the recreation benefit. This is because the increased property value calculated in the previous section already included the recreational value for houses in the radius of 200 m around Thinking Fadura.

Zones excluded in the estimation of the recreation benefit to avoid double counting with the increased property value. For public transport users, it was estimated from the survey to visitors of the park that the mean time spent was 0.4 h.

##### Physical Activity

Although there is no solid evidence base as yet, there are some studies that indicate that urban greenspaces such as parks, woodlands and beaches have the potential to support and encourage regular outdoor physical activity [45,46]. Green spaces offer opportunities for physical activity among people who, due to lack of time, money or confidence, are reluctant to participate in organised sports or gym-related activities [45,47,48]. As Thinking Fadura restored a pathway and provided access to a green space to the general public, it could be argued that some people might use the pathway for doing physical exercise such as jogging, walking or cycling (Equation (7)):(7)PA =PIPA×TPA×EQ× QV
where PA is the physical activity value, *PIPA* is the number of people increasing physical activity, *TPA* is the time spent doing physical activity, *EQ* is the effect on QALY ($\frac{\min}{\mathrm{person}}$) and QV ($\frac{QALY}{\min}$).

The mean number of people increasing physical exercise per year was calculated through the iSOPARC tool ([32], see the description of the iSOPARC assessment in Thinking Fadura in the Appendix A). Only daylight time during non-rain events was included in the analysis. For doing so, the mean rain time during the year was calculated from the closest weather station in Getxo because at other times the numbers of users in the park was considered as negligible.

Subsequently, the potential health effect associated with the cumulative level of physical activity was estimated in terms of Quality Adjusted Life Years (QALYs). We used QALY estimates derived by [45,49] which aimed to estimate the potential health benefits of greenspaces to promote physical activity. Beale et al. [49] estimated that 30 min a week of moderate to intense physical activity during the whole year would be equivalent to 0.010677 QALYs per individual and year. It was considered that 30 min a week of physical activity would be equivalent to 1560 min a year. Then, the health effect was converted into monetary terms. Pinto Prades [50] estimated the implicit social value of a QALY in the Basque Country of €22,400 and [45] the value in England of £ 20,000 (€22,968.16 in June 2019)

#### 2.3.2. Costs

##### Initial Investment and Operation and Maintenance (O&M)

The Thinking Fadura project requires actions that included some initial investment, operation and maintenance costs in order to open the park to the general public entailing a number of improvements and restoration. These costs include the reconstruction of the roads on both banks of the river and throughout the public space, the construction of the new access to the main building and to a car park located in the central area of Fadura, the intervention in the fenced area both to eliminate the current perimeter limits and to install the new ones, the conditioning of the outer slopes in the area of the present uncovered paddle tennis court, the expenses of renovation of furniture and outdoor lighting along the river walk and in the recreation areas. In addition, the replanting of the park and the adequacy and restoration of the land is also considered in the budget.

##### Dis-Benefits

Some dis-benefits (a disadvantage or loss resulting from implementing Thinking Fadura) were considered in the CBA, including an increase in pollutant emissions and noise pollution from cars as a result of the increased number of recreational visits.

The other dis-benefit considered in the CBA was slow traffic on the road during construction works (Equation (8)). Construction works have been considered as one of the major reasons for traffic congestion [51,52]. This traffic delay represents a significant cost to the commuters and other drivers, in which time lost is the major concern.
(8)ST = TD × PC × CCW × DST × TOC
where ST indicates the value of slow traffic congestion in euros, TD indicates the average traffic delay, PC is the number of people per car, CCW is the number of affected cars during construction works, DST is the number of days of slow traffic and TOC is the time opportunity cost.

Construction works have been considered as one of the major reasons for traffic congestion [51,52]. This traffic delay represents a significant cost to the commuters and other drivers, in which time lost is the major concern. The time delay depends on a large number of variables. For instance, while construction work has been estimated to cause a traffic delay for two minutes on average in Nigeria [53], more delay time is found in the US, where in New Jersey and Salt Lake City construction causes a traffic delay of approximately 10–15 min [54]. Since the construction works associated with Thinking Fadura project were relatively small (urbanization of a small part of the road and the new access to the main building and to a car park), it was assumed that the average traffic delay would be one minute during ten days of slow traffic. The amount of traffic was counted at different times of the day during the construction period.

### 2.4. Sensitivity Analysis

Sensitivity analysis is used to assess the robustness of economic assessments by focusing on how uncertainty in the input parameters propagates throughout the overall analysis. Typically, the sources of uncertainty are derived from subjective judgements of the researcher such as assuming values of certain parameters in the CBA. In Thinking Fadura, four criteria or sources of uncertainty were evaluated in the sensitivity analysis:Discount rate: Three different discount rates were evaluated (0%, 3.5% and 10%)Time horizon: Three time horizons were included (10, 20 and 30 years)Assumed values for items with high uncertainty: These parameters are reported in Table 2. Three scenarios were included: −20%, 0% and +20% of the assumed value of the items.Changes in future use: Since the greenspaces were only partially opened at the time when the survey was implemented it was considered that future use could increase over time. Throughout the survey to the general public, the percentage of respondents that would increase their frequency of use once the park is fully opened was estimated (24.8%). This criterion includes two scenarios: considering that future use will be the same, and considering that future use will increase by 24.8%.

The four criteria led to a total of 54 distinct scenarios. In these scenarios, the economic benefits and costs the NPV as well as their evolution throughout time were assessed for the sensitivity analysis.

## 3. Results

### 3.1. Participatory Evaluation

The participation of stakeholders and experts, visitors and the general public contributed to a more thorough analysis of the impacts of Thinking Fadura project. Interestingly, stakeholders and experts, visitors and the general public all had differing views of the positive and negative impacts (Table 3). In the workshop, a number of impacts not identified before such as greater influx of dogs, possible conflict between users and greater possibility of teens drinking were detected. These negative impacts were not included in the economic evaluation due to its complexity to convert into monetary terms. Nevertheless, their relative importance in comparison with the other impacts was quantitatively evaluated through 7-point Likert scales.

Whilst improved recreational activity, improved accessibility to people with lower economic status and vulnerable groups and better connected community were seen as the most beneficial impacts by respondents, more noise, dirt, and insecurity, possible conflict between users and greater possibility of teens drinking were perceived as the most negative impacts. The increase in property values around Fadura is perceived as beneficial but at the lowest range, with the general public being slightly more positive about it than the users. For certain impacts, such as higher maintenance costs, there is quite a large difference between the opinions of visitors and general public.

The potential positive impacts were rated high on the scale, likewise the negative consequences were generally unfavourable amongst both categories of respondents. The assessments show that with some exceptions, key stakeholders and experts, visitors and the general public in general had similar perceptions of the positive and negative impacts.

### 3.2. The Economic Assessment of Thinking Fadura

The CBA was conducted for a time horizon of 20 years and a 3.5% discount rate. The results demonstrate that Thinking Fadura seems beneficial with a NPV of around €1.2 million (see Table 4 and Table 5). The highest benefits are the amenity and recreational value (around €2.5 million). The highest dis-benefit was slow traffic (congestion) during construction works which was around €8000. Within the initial investment and O & M, the highest cost was land adjustment including parking (around €0.8 million). The internal rate of return (IRR) was 11.7%, the payback period 10.6 years and the benefit cost ratio around 1.63 (see Table 5).

The discounted cumulative cash flow, benefits and costs in the evaluated 54 scenarios in the sensitivity analysis are shown in Figure 2. Whilst the black curve shows the selected scenario (discount rate = 3.5%, time horizon = 20 years, assumed values for items with high uncertainty = 0%, and changes in future use = 0%), the grey area indicates the confidence intervals estimated as the 5% and 95% percentiles of the calculated values in the 54 scenarios. The graph shows that from year 13 onwards Thinking Fadura seems to present higher benefits than costs in most of the scenarios. The width of the grey area, which indicates the range of variability, increases as the time horizon increases.

Figure 3 shows the distribution of the values of the disaggregated benefits, costs and NPV (Total) in the 54 scenarios. As shown, physical activity, recreation and the increased value of property houses are the benefit items that present the highest variability depending on the chosen scenario. The high variability in these three benefits is reflected subsequently in the total NPV. The sensitivity analysis shows that the intervention is economically feasible (above 0 euros) in 52 scenarios (96.3% of the 54 scenarios). Thinking Fadura project presents a negative NPV at a 10% discount rate, time horizon of 20 years and/or 30 years, −20% of the assumed values for items with high uncertainty, and no increase in future use. For the remaining scenarios the NPV of the pilot is positive.

Overall, the sensitivity analysis shows that the highest source of uncertainty in the CBA comes from the distinct values of the discount rates (0%, 3.5% and 10%), having an inverse relationship between NPV and discount rate. Amenities and recreation (increased value of houses and recreation) and physical activity which represent the highest benefits are notably influenced by the discount rate. The second highest source of uncertainty is the time horizon followed by the parameters’ assumed values. Changes in future use had the least influence on total estimated NPV—given the importance of other factors in driving the benefits this is perhaps unsurprising. Uncertainty can also be analysed by type of benefit. For example, changes in future use affects mainly recreational benefits (after discount rate and time horizon which have the highest impact), while uncertainty in CBA parameters affects mainly the house pricing estimation, after discount rate.

## 4. Discussion

Since urban developments are often expensive investments accounting for economic benefits and costs of green spaces represents a useful tool to support policy makers and urban designers in decision-making processes. This paper explores the use of a PBCA to evaluate the benefits and costs of the Thinking Fadura project considering impacts from the point of view of society as a whole. It took into account the perspectives of all social actors or stakeholders affected by the implementation of the project itself, considering environmental and social variables that could be reasonably quantified through market and non-market valuation methods.

The PBCA of this paper has shown the synergies of combining a participatory process and a CBA. Whilst the CBA allowed comparing the NPV of different scenarios against the status-quo, the participatory process contributed to identifying and analysing people’s perceptions. The participatory evaluation improves the robustness and quality of the economic evaluation and increases social acceptability of implementation. However, the engagement process needs to be multidirectional, not only within the scale of the case study and its diverse stakeholder groups, but also on a larger scale, including promotion to the general public or the rest of the society to ensure future social acceptability [22]. In our study, one of the strengths was the fact that the evaluation was not only based on the perspective of the researchers which is the case of numerous economic assessments but also on the perspective of stakeholders, experts, visitors and the general public. This social engagement process can serve to inform future steps in the development of Thinking Fadura by the municipality of Getxo in terms of promoting future use while reducing risks and negative impacts as perceived by the population.

A key aspect of the PBCA was the involvement of the key local agents responsible for the implementation of the pilot (practice) in the evaluation process. Local agents were involved in all key methodological steps: identification of status quo and policy change, vulnerable groups, categories of impacts, data on implementation costs, time horizon for the evaluation, quantitative evaluation of key CBA parameters, survey development, and the stakeholders’ workshop. It is worth highlighting that some potential negative impacts such as teenagers gathering for drinking alcoholic drinks or greater influx of dogs were not identified by the researchers. Nevertheless, thanks to the stakeholder participation in the workshop, these impacts were included in the analysis and evaluated qualitatively and quantitatively through Likert scales. Moreover, ad hoc stakeholders’ participation processes and citizens’ surveys were put in place to gather specific information needed for the evaluation, including categories of impacts and indicators, current and future use, perceptions on impacts, attitudes, and socio-economic factors. Another key point was the analysis of the citizens’ perspective (through ad hoc surveys) to analyse perceptions and attitudes of people on the acceptability and impacts of the practice. This analysis was useful to complement the economic evaluation based on quantitative indicators.

Key benefits are related to people increasing their use of green spaces, resulting in increased recreational and amenity values, as well as improved physical activity and health. However, despite the fact that the beneficial health effects of green spaces have recently gained wide recognition, epidemiological studies sometimes have shown mixed results with significance varying considerably by study and context, indicating that there is no unique and clear evidence. Chiabai et al. (2020) found key patterns emerging throughout the literature and identified main determinants affecting the relationship between green spaces and health effects. Taking into account the correction for the publication bias, the authors found the highest health risk reductions in poorer neighbourhoods as a result of increased green space exposure.

This study combines the results of the CBA with the perceptions of citizens on impacts generated by opening Fadura park. In the survey, the majority of respondents considered as very positive impacts those related to recreational activity and improved health, trees and biodiversity, improved accessibility to vulnerable groups and a better-connected community. The increased value of house prices was considered very important by a small number of respondents and some even contemplated it as a negative impact (though a very low percent). Trees and biodiversity were not assessed, as in reality there were very small changes in this respect in comparison with the baseline scenario of the Fadura park. Improved accessibility to vulnerable groups and a better-connected community are intangible benefits which could not be converted into monetary values, though they are key factors to consider for decision-making. Intangible costs, which also cannot be translated into a monetary impact, included feeling of insecurity and increased litter, teenagers gathering for drinking, possible conflicts among users, and greater influx of dogs. Moreover, based on the membership data from the Fadura´s Municipal Sports Center, there was no significant change in the revenue from the annual membership fee as a result of the park opening. Thus neither dis-benefits nor benefits associated to changes in the revenue from the annual membership fee were considered in the CBA.

This study was implemented within the INHERIT project (www.inherit.eu) which developed a theoretical framework aimed at providing a triple win in environment, health and social equity terms when implementing policies and public investments. Thinking Fadura aimed to achieve a triple win via this municipal initiative. The economic assessment showed the triple win was achieved since removing the barriers to green space access provided health, recreation and environmental benefits and there were reductions in inequalities in terms of access. Although it was not possible to attribute monetary values to the social equity impact, this was perceived as among the highest benefits in the citizen’s survey.

The opening of green spaces which were previously not accessible to the public may create opportunities for adaptation to climate change beyond the health and recreational benefits. This is because the opening requires works for improvement of the green space before making it available to the general public (such as planting more trees), with improved provision of ecosystem services. In addition, the opening of green spaces is expected to support health benefits for the general population which become more resilient in face of climate change and global risks in general. Interventions in urban green spaces, such as increasing access for the general public are, expected to have long-term impacts, where implementation costs are supported in the present, while benefits are usually seen in the future. In these cases, it is important for public administrations to know the time period required for the social return on the investment to be sufficient. An evaluation of these interventions is necessary to analyse the economic convenience for the society as well as the social acceptability, in order to avoid situations of maladaptation or non-profitable use of public resources. Nevertheless, these adaptation measures should follow specific guidelines for the evaluation of costs and benefits (quantifiable and measurable in monetary terms through a set of methods), as well as citizens’ perceptions on expected future impacts. Both tangible impacts (monetised) and intangible impacts (not monetisable) should be compared with citizens’ perceptions to complement the CBA results and inform next phases of implementation in interim analysis. For this reason, this type of analysis, carried out at an interim phase, shows results which can be used to inform the next stages during the implementation phase. When carried out ex post, they can inform decisions on similar projects.

## 5. Conclusions

Overall, our findings suggest that removing barriers to access to green spaces may yield significant health and environmental benefits—and aid in reducing inequalities of access, while supporting adaptation to climate change. Thinking Fadura aimed at delivering “triple wins” and the results show that the investment’s benefits outweigh the costs. In the selected scenario (discount rate = 3.5%, time horizon = 20 years, assumed values for items with high uncertainty = 0%, and changes in future use = 0%), the NPV was around €1.2 million, and the payback period was 10.6 years. Starting from year 13 onwards the Thinking Fadura project presents higher cumulative benefits than costs and therefore positive NPVs in the majority of scenarios. This was in line with the existing literature, where for most projects related to green infrastructures, the discounted payback period often tends to extend to more than ten years [55,56]. The sensitivity analysis tested the significance of the variables to the NPV and to the economic value of the benefits and costs. Amenities and recreation was the most sensitive benefit to the analysed sources of uncertainty. It was not surprising that this variable has the greatest influence to the NPV as it shared the highest proportion among all the benefits in terms of present value. Most scenarios showed a positive NPV (52 over the 54 scenarios produced), a B/C ratio greater than 1, and an IRR greater than the selected discount rate (3.5%). Based on these decision rules, the Thinking Fadura project is hence considered as economically feasible from a societal perspective. Although the input data in the assessment are context specific, the Thinking Fadura project in itself could serve as a reference in the decision-making process in numerous European case studies. Firstly, there are numerous green urban areas in Europe where the use of green spaces is restricted to certain sections of the population which was the case with the sporting area of Fadura, where only members of the sporting club could use and enjoy the green spaces. Secondly, the case study of Fadura exemplifies how public sporting clubs can remove their fences and grant open access to the general public in order to increase societal usage of green spaces.

Finally, participatory Benefit-Cost analysis presents a useful technique in bringing together qualitative and quantitative estimation of the impacts of opening green spaces on societal welfare. It can help to highlight areas where there may be difficulties in the implementation process, or where there may need to be mitigation actions to reduce unintended negative impacts. It could be considered as an appropriate framework to facilitate communication among different fields of expertise and to support the identification of key indicators for future evaluations.

## Figures and Tables

**Figure 1 ijerph-17-02818-f001:**
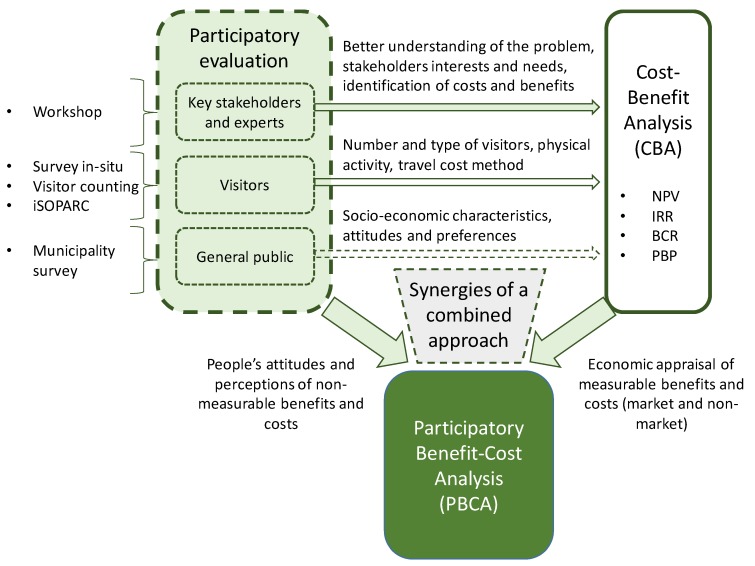
Conceptual framework to evaluate the Thinking Fadura project based on Participatory Cost Benefit Analysis. iSOPARC: System for Observing Play and Active Recreation in Communities; NPV: Net Present Value; IRR: Internal Rate of Return; BCR: Benefit-Cost Ratio; PBP: Payback Period.

**Figure 2 ijerph-17-02818-f002:**
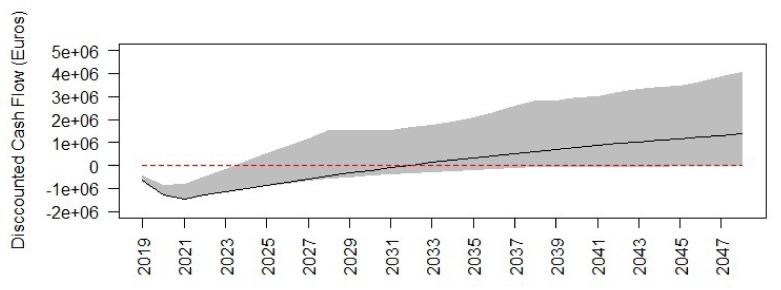
Discounted cumulative cash flow in Thinking Fadura. Black curve shows the selected scenario (discount rate = 3.5%, time horizon = 30 years, Assumed values for items with high uncertainty = 0%, and Changes in future use = 0%). Grey area indicates the range of variability. Upper limit and lower limit of the grey area indicate percentile 5 and percentile 95 of the 54 evaluated scenarios in the sensitivity analysis. Red and dashed line indicates 0 euros.

**Figure 3 ijerph-17-02818-f003:**
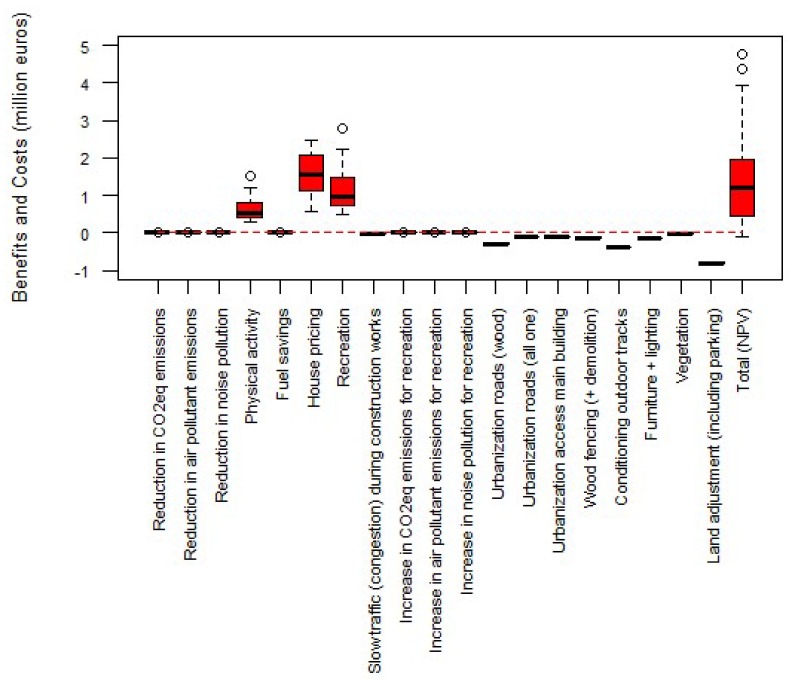
Distribution of the values of the benefits, costs and NPV in the different scenarios in Thinking Fadura project. The horizontal lines in the boxes indicate the median values (percentile 50) of the 54 scenarios. The box limits indicate percentile 25 and 75. The whiskers are calculated as 1.5 times the interquartile range (Q3–Q1). Circles indicate values out of the interquartile range (outliers). Red and dashed line indicates the value 0 euros.

**Table 1 ijerph-17-02818-t001:** Proportion of visitor’s residence distance and transport means in the greenspaces of Fadura.

Transport Means	Zone 1 ^a^ (125 m)	Zone 2 ^a^ (250 m)	Zone 3 (750 m)	Zone 4 (2000 m)	Proportion by Transport Means	Estimated Mean Distance (m)	Mean Speed (km h^−1^)
On foot	28.9%	14.8%	13.7%	25.4%	82.4%	848	5
Bicycle	2.0%	0.4%	1.2%	3.1%	6.6%	1132	10
Car or motorbike	0.4%	0.4%	1.2%	7.4%	9.4%	1698	30
Public transport	0.0%	0.0%	0.0%	1.2%	1.2%	2000	− ^b^

a: Zones excluded in the estimation of the recreation benefit to avoid double counting with the increased property value, b: For public transport users, it was estimated from the survey to visitors of the park that the mean time spent was 0.4 h.

**Table 2 ijerph-17-02818-t002:** Items with highest uncertainty in the CBA of Thinking Fadura.

Items with High Uncertainty	Assumed Values
Average reduced distance (km/car)	5
Percentage of people walking or cycling instead of driving (%)	1%
Mean time per iSOPARC scan (min/iSOPARC scan)	10
Number of days of slow traffic (days)	10
Average traffic delay (h)	0.0167
Mean amount of time doing physical exercise (min/person in each visit)	10
Price increase of houses (%)	1%

**Table 3 ijerph-17-02818-t003:** Perceptions of the positive and negative impacts of implementing the Thinking Fadura project from the key stakeholders and experts of the workshop, visitors of the in-situ survey, and Getxo citizens of the general public survey. Figures indicate mean values in a Likert scale from −10 to +10 and relative change (%). Green and red colors indicate higher and lower values respectively.

Items	Key Stakeholders and Experts ^1^	Visitors ^2^	General Public ^2^	Relative Change between Visitors and General Public
		Mean (SD)	Mean (SD)	
Improved recreational activity	2.52 (*n* = 5)	2.22 (0.98)	2.59 (0.83)	−14.30%
Improvement of physical and mental health of green zone users	NA (*n* = 0)	2.45 (0.80)	2.56 (0.75)	−4.30%
More trees and biodiversity	NA (*n* = 0)	2.60 (0.67)	2.77 (0.53)	−6.10%
Provide access to people with lower economic status and vulnerable groups	3 (*n* = 1)	2.47 (0.78)	2.62 (0.80)	−5.70%
Better connected community	1.93 (*n* = 7)	2.42 (0.84)	2.65 (0.76)	−8.70%
More noise, dirt, and insecurity	−3 (*n* = 2)	−1.32 (1.11)	−1.65 (1.27)	20.00%
Greater influx of dogs	−1.8 (*n* = 5)	0.22 (1.85)	0.42 (2.00)	−47.60%
Higher maintenance costs	NA (*n* = 0)	0.03 (1.48)	−0.54 (1.91)	105.60%
Possible conflict between users	−1.8 (*n* = 3)	−0.90 (1.01)	−0.94 (1.21)	4.30%
Greater possibility of teens drinking	−1.95 (*n* = 8)	−1.52 (1.19)	−1.77 (1.23)	14.10%
Increase of value of houses around Fadura	NA (*n* = 0)	0.65 (1.41)	1.68 (1.38)	−61.30%

(1) Responses from the workshop. The number of observations (n) varies across the items because stakeholders and experts were asked to identify themselves the impacts of opening to the general public the greenspaces of Fadura. Subsequently, they were asked to quantified the impacts in a Likert scale. Values in brackets indicate the number of observations (n). (2) Responses from the visitors’ survey (n = 256) and from the general public survey (n = 250). In both surveys, respondents responded to the question “How would you assess the impacts that could arise as a consequence of opening the park?” (−3 meaning very negative, 0 neutral, and +3 meaning very positive). Values in brackets indicate the standard deviation.

**Table 4 ijerph-17-02818-t004:** Summary of Benefits and Costs of Thinking Fadura project. Time horizon 20 years and discount rate 3.5%.

Benefits and Costs	Group	Items	Items (€)	Group (€)
Benefits	Environment and Health	Reduction in CO_2_eq emissions	62 €	597,260 €
		Reduction in air pollutant emissions	119 €	
		Reduction in noise pollution	46 €	
		Physical activity	597,033 €	
	Economics	Fuel savings	3113 €	3113 €
	Amenity and Recreational value	Amenity and Recreational value	2,484,926 €	2,484,926 €
Costs	Investment and O & M	Restructuring roads (wooden part)	−286,103 €	−1,959,591 €
		Restructuring roads	−96,656 €	
		Reconstruction access main building	−98,590 €	
		Wood fencing (+ demolition)	−127,587 €	
		Conditioning outdoor tracks	−386,626 €	
		Furniture + lighting	−140,091 €	
		Vegetation	−19,662 €	
		Land adjustment (including parking)	−804,277 €	
	Dis-benefits	Slow traffic (congestion) during construction works	−8187 €	−8671 €
		Increase in CO_2_eq emissions due to recreation	−131 €	
		Increase in air pollutant emissions due to recreation	−254 €	
		Increase in noise pollution due to recreation	−99 €	
Total			1,117,044 €	1,117,044 €

**Table 5 ijerph-17-02818-t005:** Economic assessment of Thinking Fadura. Time horizon 20 years and discount rate 3.5%.

Items	Estimated Value (€) in the Selected Scenario	Estimated Value (€) in the Scenario Considering A Future Increase Use by 25%
Discounted Benefits (€)	3,085,298 €	3,476,961 €
Discounted Costs (€)	1,968,255 €	1,968,373 €
NPV (€)	1,117,044 €	1,508,588 €
Ratio B/C	1.57	1.77
IRR (%)	10.82%	13.34%
Payback Period (years)	11.10	9.62

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
