# Peer review of "Providing Access to Urban Green Spaces: A Participatory Benefit-Cost Analysis in Spain"

_ijerph, 2020, doi:10.3390/ijerph17082818_

Round 1

Reviewer 1 Report

Comments and Suggestions for Authors

The manuscript "The economic profitability of providing access to urban greenspaces: a case-study in Spain" analyzes the advantages deriving from the accessibility of citizens to green spaces. The work is well written and well structured and can be useful for other realities with the same characteristics.

Please note: the numbering of the staves is not included in the manuscript, so this is a problem for reviewers. Despite this, I tried to give some suggestions, but in this way I cannot provide many details.

The study area is not sufficiently treated. What is the context in which it fits? How many inhabitants does the municipality have? Furthermore, at an ecological-environmental level what are the characteristics of this park? What environmental impacts will the implementation of the new project have? Would the opening to everyone mean a greater turnout and consequently a greater number of machines and vehicles in general? Therefore also a greater acoustic impact and so on.

I think the introduction should include more references and paragraph 3.2. "Participatory evaluation" should be better explained by integrating more aspects of the supplementary material. This could be useful for a greater understanding for readers.
Try to briefly explain the "System for Observing Play and Active Recreation in Communities" used. Furthermore, I believe it would be useful to add the way in which the residents of the Municipality of Getxo were interviewed face to face. Did this sample also include children or teenagers?

3.3.1.5 Physical activity
Again, the iSOPARC assessment must be explained because it's information that cannot be taken for granted.

Author Response

Comments and Suggestions for Authors
1. The manuscript "The economic profitability of providing access to urban greenspaces: a casestudy in Spain" analyzes the advantages deriving from the accessibility of citizens to green
spaces. The work is well written and well structured and can be useful for other realities with
the same characteristics.
R: The authors are very grateful for the valuable comments of the reviewer.
2. Please note: the numbering of the staves is not included in the manuscript, so this is a problem
for reviewers. Despite this, I tried to give some suggestions, but in this way I cannot provide
many details.
R: The authors apologise for this. We have now included line numbers in the submitted manuscript.
3. The study area is not sufficiently treated. What is the context in which it fits? How many
inhabitants does the municipality have? Furthermore, at an ecological-environmental level
what are the characteristics of this park? What environmental impacts will the
implementation of the new project have? Would the opening to everyone mean a greater
turnout and consequently a greater number of machines and vehicles in general? Therefore
also a greater acoustic impact and so on.
R: The following text has been inserted to better describe the issues suggested by the reviewer:
“Getxo is a coastal town and had 77,946 inhabitants in 2019. Getxo is mostly an affluent residential
area, as well as being the third largest municipality of Biscay province. In regards to the ecological
characteristics of the area, Fadura is located in a very important strategic position for Getxo. The park´s
area, works like a hinge between the dense urban district, the coast, the Gobela river, and the natural
2
environment surrounding it (the Bolue wetland). It is essential to highlight the role that the river
Gobela and the Bolue wetland have in Getxo, both strictly related with Fadura, as biodiversity reserves
and possible axes for a greener city. The geographical location of the sports centre of Fadura has a
strong impact on its character, not only at the environmental and natural level but also, at the level of
human geography, urban planning and accessibility. The redesign of the park is a unique opportunity
to diversify the activities that can be carried out in this natural environment, maintaining the
continuity of the ecosystem along the ecological corridor that forms the river and trying to reach a
sound balance between accessibility for leisure and conservation of the ecosystem, respecting the
native fauna and flora.”
In regards to the last question comment (“Would the opening to everyone mean a greater turnout
and consequently a greater number of machines and vehicles in general? Therefore also a greater
acoustic impact and so on."), the following text is included in the description of the dis-benefits
(section 3.3.3.2.):
“Some dis-benefits (a disadvantage or loss resulting from implementing Thinking Fadura) were
considered in the CBA, including an increase in pollutant emissions and noise pollution from cars as a
result of the increased number of recreational visits.”
4. I think the introduction should include more references and paragraph 3.2. "Participatory
evaluation" should be better explained by integrating more aspects of the supplementary
material. This could be useful for a greater understanding for readers.
Try to briefly explain the "System for Observing Play and Active Recreation in Communities"
used. Furthermore, I believe it would be useful to add the way in which the residents of the
Municipality of Getxo were interviewed face to face. Did this sample also include children or
teenagers?
R: The introduction section has been considerably improved. See below.
1. “Introduction
The proportion of people living in urban areas is expected to rise from 46.6% in 2000 to 69.6% in 2050
[1]. As a result of this increase in urbanisation, greenspace in urban areas is facing increasing pressure.
In this context, there are significant challenges faced by planners and local authorities in the balancing
of the demands for space and development with the needs of the population for greenspaces.
To counteract the negative effects of increased urbanization, movements such as “Healthy Cities”
started by the World Health Organization have taken root in many urban areas. This movement is a
long-term international development initiative that promotes physical and social environments by
putting health high on decision makers’ priorities [2]. Part of improving accessibility to healthy
environments is to promote healthy living, which includes physical and mental well-being, as well as
increased access to spaces that promote physical activity. Greenspaces are connected to psychological
and physical benefits through their use as spaces for physical activity and mental restoration [3-5].
Urban greenspaces can provide multiple benefits to human health by fostering an increase in physical
activity [6-7], improving air quality [8], reducing noise pollution [9], and protecting against high
temperature and heavy rainfall [10]. Apart from physical health, urban greenspaces contribute to
people’s wellbeing and mental health by increasing recreation activities [11], reducing stress [5] and
stimulating social contacts and increasing societal cohesion [12].
3
In terms of climate change mitigation and adaptation, urban greenspaces play an important role first
by sequestering carbon and reducing atmospheric CO2 concentrations and then by lowering the
higher temperatures that may be accentuated by the “urban heat island” effect. This phenomenon is
when urban zones tend to be hotter on warm days than their surrounding areas [13]. One of the
reasons this may result in urban areas is the lack of vegetation to shade and cool the air.
The Inherit project (www.inherit.eu) investigated whether investing in making urban greenspaces
accessible to all has potential to contribute to a triple win of improving health, improving equity and
contributing to environmental sustainability. Bell et al. [6] described important lessons for good
practice in changing contexts to modify behaviours and achieve this triple win. Aligned with Kruize et
al. [12], they highlighted the contribution of greenspaces to well-being, and the fact that creating more
readily accessible and good quality greenspace often provide opportunities to socialise and to be
active, as well as contributing to mitigate climate change impacts. A part from having a beneficial
mental health effect, greenspaces can moderate the association of psychological distress with arealevel socio-economic status. Sugiyama et al. [5] found that the odds of higher psychological distress
was higher in residents in lower socio-economic status areas than those in higher socio-economic
status areas.
Economic analyses can be used to demonstrate quantitatively the costs and benefits of greenspaces
to policy makers and society. Municipal budgets for greenspace management are often under
pressure and consequently, there is an urgent need for new funding models to demonstrate the need
for urban greenspaces. Inspired by the TEEB study [14], van Zoest and Hopman [15] described an
initiative to develop a tool that makes the financial benefits of greenspaces visible in the municipal
balance sheet. Harnik and Welle [16] set forth a methodology for valuing the benefits of urban parks.
They found seven major factors that could be quantified and valued in parks: property value, tourism,
direct use, health, community cohesion, clean water, and clean air. [17-18] estimated the non-market
benefits derived from the provision of a new urban park where there was an old train station. Through
a contingent valuation method, they found that the mean willingness to pay was considerably higher
for people who live closer to the planned park as it was more accessible to them. Cheng and Wang
[19] showed the economic benefits of greenspaces by empirically assessing the evolving path of urban
greenspaces along with economic development and urbanization through panel data across 285
Chinese cities. The authors found an N-shaped environmental Kuznets curve, indicating that
greenspace coverage increased at the initial stage of economic development, and then it started to
decrease as GDP per capita exceeded a certain threshold and then increased again at a high GDP per
capita level.
Economic evaluations through cost-benefit analyses (CBA) typically estimate the total value of a
particular greenspace (e.g. [16-18]) and do not focus on their specific attributes such as the value of
improved accessibility or extended exposure to greenspaces. As far as the authors are aware of there
are no studies in the scientific literature that economically assess the benefits of providing or
improving greenspace access that previously was restricted to the general public. This paper addresses
this gap in the literature, by evaluating the economic benefits and costs of a project called Thinking
Fadura aimed at increasing access to restricted greenspaces. The other main literature gap that is
addressed in this study is the direct participation of key stakeholders, users of the greenspaces and
the general public in the processes for designing and implementing the CBA. Previous studies have
tried to incorporate stakeholders’ views in similar methodologies such as Multi-Criteria Decision
Making [20]. Our assessment incorporates a participatory process into a CBA and shows the synergies
of combining them for future economic evaluations. Similar approaches based on the participation of
stakeholders have been used to allow communities to compare the costs and benefits of public
interventions and/or identify disaster risk reduction measures (e.g. [21]). Campos et al. [22] found that
a multidirectional engagement process with diverse stakeholder groups and at various scales
(stakeholders, users and general public) ensures future social and sustainable acceptability. Thus the
4
combination of participatory process and a CBA enhances the robustness of the results and their social
acceptability.”
The following references have been added:
Bell, R., Khan, M., Romeo-Velilla, M., Stegeman, I., Godfrey, A., Taylor, T., Morris, G., Staatsen, B., van
der Vliet, N., Kruize, H., Anthun, K.S., Lillefjell, M., Espnes, G.A., Chiabai, A., García de Jalón, S.,
Quiroga, S., Martinez-Juarez, P., Máca, V., Zvěřinová, I., Ščasný, M., Marques, S., Craveiro, D.,
Westerink, J., Spelt, H., Karnaki, P., Strube, R., Merritt, A.S., Friberg, M., Bélorgey, N., Vos, M.,
Gjorgjev, D., Upelniece, I., Costongs, C., 2019. Ten Lessons for Good Practice for the INHERIT
Triple Win: Health, Equity, and Environmental Sustainability. Int. J. Environ. Res. Public Health
16, 4546. DOI:10.3390/ijerph16224546.
Chen, W.Y., Wang, D.T., 2013. Economic development and natural amenity: An econometric analysis
of urban green spaces in China. Urban Forestry & Urban Greening 12 (4), 435-442.
Chiabai, A., Quiroga, S., Martinez Juarez, P., Suarez, C., García de Jalón, S., Taylor, T., 2020. Exposure
to green areas: Modelling health benefits in a context of study heterogeneity. Ecological
Economics 167, 106401.
Kruize, H., van der Vliet, N., Staatsen, B., Bell, R., Chiabai, A., Muiños, G., Higgins, S., Quiroga, S.,
Martinez-Juarez, P., Aberg Yngwe, M., Tsichlas, F., Karnaki, P., Lima, M.L., García de Jalón, S.,
Khan, M., Morris, G., Stegeman, I., 2019. Urban Green Space: Creating a Triple Win for
Environmental Sustainability, Health, and Health Equity through Behavior Change. Int. J.
Environ. Res. Public Health 16, 4403.
Sugiyama, T., Villanueva, K., Knuiman, M., Francis, J., Foster, S., Wood, L., Giles-Corti, B., 2016. Can
neighborhood green space mitigate health inequalities? A study of socio-economic status and
mental health. Health & Place 38, 16-21.
TEEB, 2010. The Economics of Ecosystems and Biodiversity Ecological and Economic Foundations.
Edited by Pushpam Kumar. Earthscan, London and Washington.
van Zoest, J., Hopman, M., 2014. Taking the economic benefits of green space into account: The
story of the Dutch TEEB for Cities project. Urban Climate 7, 107-114.
Vivanco-Hidalgo, R.M., Avellaneda-Gómeza, C., Dadvand, P., Cirach, M., Oisa, Á., Gómez
González, A., Rodriguez-Campello, A., de Ceballos, P., Basagaña, X., Zabalza, A., CuadradoGodia, E., Sunyer, J., Roquer, J., Wellenius, G.A., 2019. Association of residential air pollution,
noise, and greenspace with initial ischemic stroke severity. Environmental Research 179,
108725.
García de Jalón, S., Burgess, P.J., Curiel Yuste, J., Moreno, G., Graves, A., Palma, J.H.N., CrousDurán, J., Kay, S., Chiabai, A., (2019) Dry deposition of air pollutants on trees at regional scale:
a case study in the Basque Country. Agricultural and Forest Meteorology 278, 107648.
Li, Z., Chow, D.H.C., Yao, J., Zheng, X., Zhao, W., 2019. The effectiveness of adding horizontal greening
and vertical greening to courtyard areas of existing buildings in the hot summer cold winter
region of China: A case study for Ningbo. Energy and Buildings 1961, 227-239.
Garrod, G.D., Willis, K.G., 1992. Valuing Goods' Characteristics: an Application of the Hedonic Price
Method to Environmental Attributes. Journal of Environmental Management 34, 59-76.
5
Section 3.2 has been thoroughly reviewed and improved and the following text has been added to
clarify the use of "System for Observing Play and Active Recreation in Communities":
“In Thinking Fadura, the iSOPARC assessment was carried out in order to determine the number of
people (including all ages) increasing physical exercise as a result of the restoration of the pathway
and the improved accessibility in Fadura.
The green areas of Fadura occupy around 4ha and was divided in 5 target areas (see Figure S.6 in the
Supplementary Material). The observations were done by two observers in August 2018. In total
sixteen site visits were done in each Target Area (5 TAs) at different time of the day. These times were
at 9.00am, 12.00pm, 5.00pm and 8.00pm. The site visits were done in twelve different days covering
both weekdays and weekends. Table S.2 shows the temporal distribution at different times. Every site
visit covered the five target areas.
In the five target areas, the iSOPARC tool counted an average of 2.05 people per scan. For the physical
exercise benefit only those people who were doing walking or vigorous activities were considered.
Thus in the five target areas, the average of people increasing physical exercise for walking and
vigorous activities were 1.06 and 0.21, respectively (Table S.3). Hence, the sum of people increasing
physical exercise was 1.27 (people / iSOPARC scan).”
5. 3.3.1.5 Physical activity
Again, the iSOPARC assessment must be explained because it's information that cannot be
taken for granted.
R: The following information regarding the iSOPARC assessment has been included in section 3.2.2.3
Participatory evaluation to better explain the iSOPARC tool:
“The assessment of physical activity was done with the System for Observing Play and Active
Recreation in Communities (iSOPARC, [32]) tool (see the SOPARC assessment in the Supplementary
Material). iSOPARC is an established and validated observation tool to assess the use of public spaces
in community settings [32]. iSOPARC is used to record individual (gender, age, physical activity level,
and ethnicity) and contextual characteristics (in a given area) and primary activity for each observed
person. Through visual scans, iSOPARC determines the usage and physical activity of the greenspace
at a determined time.
In Thinking Fadura, the iSOPARC assessment was carried out in order to determine the number of
people (all ages) increasing physical exercise as a result of the restoration of the pathway and the
improved accessibility in Fadura.
The green areas of Fadura (around 4ha) were divided in 5 target areas (see Figure S.6 in the
Supplementary Material). The observations were done by two observers in August 2018. In total
sixteen site visits were done in each Target Area (5 TAs) at different time of the day. These times were
at 9.00am, 12.00pm, 5.00pm and 8.00pm. The site visits were done in twelve different days covering
both weekdays and weekends. Table S.2 in the Supplementary Material shows the temporal
distribution at different times. Every site visit covered the five target areas.
In the five target areas, the iSOPARC tool counted an average of 2.05 people per scan. For the physical
exercise benefit only those people who were doing walking or vigorous activities were considered.
Thus in the five target areas, the average of people increasing physical exercise for walking and
vigorous activities were 1.06 and 0.21, respectively (Table S.3). Hence, the sum of people increasing
physical exercise was 1.27 (people / iSOPARC scan).”

Reviewer 2 Report

Comments and Suggestions for Authors

The manuscript needs a thorough review as the ideas presented do not come out clearly. The data collection methods are not explained in the Methodology; only the analytical methods are presented.

This makes it difficult to draw a connection between the data collection methods and the analytical methods employed and whether they are both appropriate.

In addition, the discussion is weak and does not provide a clear connection to the broader literature and the gaps that this research helps to fill. This stems from the fact that the introduction does not provide an indepth background to the body of knowledge that informs this research and how this research contributes to it. 

I will also encourage the author(s) to separate the discussion and conclusion section and identify critical implications for policymaking.

Author Response

Comments and Suggestions for Authors
6. The manuscript needs a thorough review as the ideas presented do not come out clearly. The
data collection methods are not explained in the Methodology; only the analytical methods
are presented. This makes it difficult to draw a connection between the data collection
methods and the analytical methods employed and whether they are both appropriate.
R: The authors appreciate the helpful comments of the Reviewer. Before the data collection methods
were described in the Supplementary Material. Now the data collection methods have been described
in detail in section 3.2 Participatory evaluation and data collection methods. See below new section:
4.1. “Participatory evaluation and data collection methods
The participatory evaluation followed a bottom-up approach in a sequential evaluation including three
main focal points of the analysis: key stakeholders and experts, visitors and the general public. This
assessment may provide a better understanding of the needs of implementing Thinking Fadura and
their impacts. The participation of stakeholders and experts, visitors and the general public also helped
to design a more precise economic evaluation.
4.1.1. Key stakeholders and experts
The first step in the PBCA was the assessment with key stakeholders and experts. A workshop with 20
participants was held in May 2018, including members from public services of different areas such as
social welfare, equality, multiculturalism, development cooperation, environment, urban planning,
housing, civil protection, economic promotion and health and Fadura sporting area. In the
development of the workshop there were two people facilitating the session.
7
The workshop had four objectives: i) to provide a better understanding of the context of Thinking
Fadura, ii) to identify the elements and the population affected by opening the greenspaces, iii) to
identify new uses in the greenspaces and relate them with potential impacts and iv) to assess the
potential impacts as a result of increasing accessibility in the greenspaces (more details in the
description of the workshop in the Supplementary Material).
4.1.2. Visitors
Subsequently, an assessment focusing on users of the greenspaces of Fadura was carried out between
August 2018 and June 2019. This included counting the users, an in-situ survey to visitors and an
assessment of physical activity.
4.1.2.1. Counting the users
The counting of visitors was done through manual counters at different times of the day, days of the
week and months of the year. The counting of visitors was needed for quantifying the recreation
benefit by the travel cost method [31].
4.1.2.2. In-situ survey to visitors
The in-situ survey to visitors provided quantitative data on frequency of use and potential increase in
the use of the park, travel cost values, and perceptions about the benefits and costs of Thinking Fadura
(for more detail see the Supplementary Material). Through this survey we obtained both qualitative
and quantitative information that was used for the different analysis carried out in this study.
The target population was adults who transited through the green areas that were currently open to
the public. A total of 256 visitors over 16 years old of age were surveyed.
4.1.2.3. Physical activity assessment
The assessment of physical activity was done with the System for Observing Play and Active Recreation
in Communities (iSOPARC, [32]) tool (see the SOPARC assessment in the Supplementary Material).
iSOPARC is an established and validated observation tool to assess the use of public spaces in
community settings [32]. iSOPARC is used to record individual (gender, age, physical activity level, and
ethnicity) and contextual characteristics (in a given area) and primary activity for each observed
person. Through visual scans, iSOPARC determines the usage and physical activity of the greenspace
at a determined time.
In Thinking Fadura, the iSOPARC assessment was carried out in order to determine the number of
people (all ages) increasing physical exercise as a result of the restoration of the pathway and the
improved accessibility in Fadura.
The green areas of Fadura (around 4ha) were divided in 5 target areas (see Figure S.6 in the
Supplementary Material). The observations were done by two observers in August 2018. In total
sixteen site visits were done in each Target Area (5 TAs) at different time of the day. These times were
at 9.00am, 12.00pm, 5.00pm and 8.00pm. The site visits were done in twelve different days covering
both weekdays and weekends. Table S.2 in the Supplementary Material shows the temporal
distribution at different times. Every site visit covered the five target areas.
In the five target areas, the iSOPARC tool counted an average of 2.05 people per scan. For the physical
exercise benefit only those people who were doing walking or vigorous activities were considered.
Thus in the five target areas, the average of people increasing physical exercise for walking and
vigorous activities were 1.06 and 0.21, respectively (Table S.3). Hence, the sum of people increasing
physical exercise was 1.27 (people / iSOPARC scan).
8
4.1.3. General public
The final step in the PBCA was the assessment of opinions among the general public. This was done
through a face-to-face survey in the municipality of Getxo (see the description of the survey to the
general public in the Supplementary Material). The aim of this survey was to study the opinion of the
citizens of Getxo with regards to Thinking Fadura´s initiative of opening Fadura´s greenspaces to the
general public. The target population was people over 16 years old and registered as citizens of the
municipality of Getxo. To identify this population, two screening questions were included in the
introduction of the survey. In total, 250 surveys were conducted in 12 different areas of the
municipality.”
7. In addition, the discussion is weak and does not provide a clear connection to the broader
literature and the gaps that this research helps to fill. This stems from the fact that the
introduction does not provide an in depth background to the body of knowledge that informs
this research and how this research contributes to it.
R: The introduction has been carefully reviewed and improved (see response to comment #4). The
contribution of the manuscript to the literature has been clarified as well as the research gaps covered
by this study. The literature review and references in the introduction has been enlarged.
The discussion section has been considerably improved. See below the new section:
5. “Discussion
Since urban developments are often expensive investments accounting for economic benefits and
costs of greenspaces represents a useful tool to support policy makers and urban designers in
decision-making processes. This paper explores the use of a PBCA to evaluate the benefits and costs
of the Thinking Fadura project considering impacts from the point of view of society as a whole. It took
into account the perspectives of all social actors or stakeholders affected by the implementation of
the project itself, considering environmental and social variables that could be reasonably quantified
through market and non-market valuation methods.
The PBCA of this paper has shown the synergies of combining a participatory process and a CBA. Whilst
the CBA allowed comparing the NPV of different scenarios against the status-quo, the participatory
process contributed to identifying and analysing people’s perceptions. The participatory evaluation
improves the robustness and quality of the economic evaluation and increases social acceptability of
implementation. However, the engagement process needs to be multidirectional, not only within the
scale of the case study and its diverse stakeholder groups, but also on a larger scale, including
promotion to the general public or the rest of the society to ensure future social acceptability [22]. In
our study, one of the strengths was the fact that the evaluation was not only based on the perspective
of the researchers which is the case of numerous economic assessments but also on the perspective
of stakeholders, experts, visitors and the general public. This social engagement process can serve to
inform future steps in the development of Thinking Fadura by the municipality of Getxo in terms of
promoting future use while reducing risks and negative impacts as perceived by the population.
A key aspect of the PBCA was the involvement of the key local agents responsible for the
implementation of the pilot (practice) in the evaluation process. Local agents were involved in all key
methodological steps: identification of status quo and policy change, vulnerable groups, categories
of impacts, data on implementation costs, time horizon for the evaluation, quantitative evaluation of
9
key CBA parameters, survey development, and the stakeholders’ workshop. It is worth highlighting
that some potential negative impacts such as teenagers gathering for drinking alcoholic drinks or
greater influx of dogs were not identified by the researchers. Nevertheless, thanks to the stakeholder
participation in the workshop, these impacts were included in the analysis and evaluated qualitatively
and quantitatively through Likert scales. Moreover, ad hoc stakeholders’ participation processes and
citizens’ surveys were put in place to gather specific information needed for the evaluation, including
categories of impacts and indicators, current and future use, perceptions on impacts, attitudes, and
socio-economic factors. Another key point was the analysis of the citizens’ perspective (through ad
hoc surveys) to analyse perceptions and attitudes of people on the acceptability and impacts of the
practice. This analysis was useful to complement the economic evaluation based on quantitative
indicators.
Key benefits are related to people increasing the use of greenspaces, resulting in increased
recreational and amenity values, as well as improved physical activity and health. However, despite
the fact that the beneficial health effects of greenspaces have recently gained wide recognition,
epidemiological studies sometimes have shown mixed results with significance varying considerably
by study and context, indicating that there is no unique and clear evidence. Chiabai et al. (2020) found
key patterns emerging throughout the literature and identified main determinants affecting the
relationship between greenspaces and health effects. Taking into account the correction for the
publication bias, the authors found the highest health risk reductions in poorer neighbourhoods as a
result of increased greenspace exposure.
This study combines the results of the CBA with the perceptions of citizens on impacts generated by
opening Fadura park. In the survey, the majority of respondents considered as very positive impacts
those related to recreational activity and improved health, trees and biodiversity, improved
accessibility to vulnerable groups and a better-connected community. The increased value of house
prices was considered very important by a small number of respondents and some even contemplated
it as a negative impact (though a very low percent). Trees and biodiversity were not assessed, as in
reality there were very small changes in this respect in comparison with the baseline scenario of the
Fadura park. Improved accessibility to vulnerable groups and a better-connected community are
intangible benefits which could not be converted into monetary values, though they are key factors
to consider for decision-making. Intangible costs, which also cannot be translated into a monetary
impact, included feeling of insecurity and increased litter, teenagers gathering for drinking, possible
conflicts among users, and greater influx of dogs. Moreover, based on the membership data from the
Fadura´s Municipal Sports Center, there was no significant change in the revenue from the annual
membership fee as a result of the park opening. Thus neither dis-benefits nor benefits associated to
changes in the revenue from the annual membership fee were considered in the CBA.
This study was implemented within the Inherit project (www.inherit.eu) which developed a
theoretical framework aimed at providing a triple win in environment, health and social equity terms
when implementing policies and public investments. Thinking Fadura aimed to achieve a tripe win in
order to achieve people’s usage and engagement with municipal initiative. The economic assessment
showed the triple win was achieved since removing the barriers to greenspace access provided health,
recreation and environmental benefits and there were reductions in inequalities in terms of access.
Although it was not possible to attribute monetary values to the social equity impact, this was
perceived as among the highest benefits in the citizen’s survey.
The opening of greenspaces which were previously not accessible to the public may create
opportunities for adaptation to climate change beyond the health and recreational benefits. This is
because the opening requires works for improvement of the green space before making it available
to the general public (such as planting more trees), with improved provision of ecosystem services. In
addition, the opening of greenspaces is expected to support health benefits for the general population
10
which become more resilient in face of climate change and global risks in general. Interventions in
urban greenspaces such increasing access for the general public are expected to have long-term
impacts, where implementation costs are supported in the present, while benefits are usually seen in
the future. In these cases, it is important for public administrations to know the time period required
for the social return on the investment to be sufficient. An evaluation of these interventions is
necessary to analyse the economic convenience for the society as well as the social acceptability, in
order to avoid situations of maladaptation or non-profitable use of public resources. Nevertheless,
these adaptation measures should follow specific guidelines for the evaluation of costs and benefits
(quantifiable and measurable in monetary terms through a set of methods), as well as citizens’
perceptions on expected future impacts. Both tangible impacts (monetised) and intangible impacts
(not monetisable) should be compared with citizens’ perceptions to complement the CBA results and
inform next phases of implementation in interim analysis. For this reason, this type of analyses, carried
out in interim, shows results which can be used to inform next stages during the implementation
phase. When carried out ex post, they can inform decisions on similar projects.”
8. I will also encourage the author(s) to separate the discussion and conclusion section and
identify critical implications for policymaking.
R: Discussions and Conclusions section has now been divided into two separate sections. Critical
implications for policymaking have been included in the discussion (see answer to comment #7).
See below the new conclusion section:
6. “Conclusions
Overall, our findings suggest that removing barriers to access greenspaces may yield significant health
and environmental benefits – and aid in reducing inequalities of access, while supporting adaptation
to climate change. Thinking Fadura aimed at delivering “triple wins” and the results show that the
investment’s benefits outweigh the costs. In the selected scenario (discount rate = 3.5%, time horizon
= 20 years, assumed values for items with high uncertainty = 0%, and changes in future use = 0%), the
NPV was around € 1.2 million, and the payback period was 10.6 years. Starting from year 13 onwards
Thinking Fadura project presents higher cumulative benefits than costs and therefore positive NPVs
in the majority of scenarios. This was in line with the existing literature, where for most projects
related to green infrastructures, the discounted payback period often tends to extend to more than
ten years [55-56]. The sensitivity analysis tested the significance of the variables to the NPV and to the
economic value of the benefits and costs. Amenities and recreation was the most sensitive benefit to
the analysed sources of uncertainty. It was not surprising that this variable has the greatest influence
to the NPV as it shared the highest proportion among all the benefits in terms of present value. Most
scenarios showed a positive NPV (52 over the 54 scenarios produced), a B/C ratio greater than 1, and
an IRR greater than the selected discount rate (3.5%). Based on these decision rules, the Thinking
Fadura project is hence considered as economically feasible from a societal perspective. Although the
input data in the assessment are context specific, the Thinking Fadura project in itself could serve as
a reference in the decision-making process in numerous European case studies. Firstly, there are
numerous green urban areas in Europe where the use of greenspaces is restricted to certain sections
of the population which was the case of the sporting area of Fadura, where only members of the
sporting club could use and enjoy the greenspaces. Secondly, the case study of Fadura exemplifies
how public sporting clubs can remove their fences and grant open access to the general public in order
to increase societal usage of greenspaces.
11
Finally, participatory Benefit-Cost analysis presents a useful technique in bringing together qualitative
and quantitative estimation of the impacts of opening greenspaces on societal welfare. It can help to
highlight areas where there may be difficulties in the implementation process, or where there may
need to be mitigation actions to reduce unintended negative impacts. It could be considered as an
appropriate framework to facilitate communication among different fields of expertise and to support
the identification of key indicators for future evaluations.”

Reviewer 3 Report

Comments and Suggestions for Authors

Using the Thinking Fadura project as a case study, this paper attempts to assess the societal profitability of removing barriers to access to urban greenspaces. The framework of Participatory Benefit-Cost Analysis (PBCA) was developed to analyze the benefits and costs of greenspaces, and multiple scenarios were designed to analyze the sensitivity of the results. This study is interesting and I think it can contribute in literature. There are some suggestions for further improvement.

Authors should add a subsection to clarify the key data sources used in this paper. How were the participatory evaluation of stakeholders and experts, visitors and the general public aided the traditional CBA? This paper showed only two separate parts. Some items of participatory evaluation were not involved in CBA. Why? The basis for selecting the respondents of this study should be further explained. Namely, the representativeness of samples. Authors need to provide more explanations for their designed items of participatory evaluation. And how these items aid the CBA. In the “3.4. Sensitivity analysis” subsection, all these 54 scenarios were analyzed using the same basic surveys. What does this effect the analysis results? In different situations, will the perception of the respondents not change? Please explain it. 6.The terms in the paper (both text, tables and figures) should be consistent, e.g. greenspaces, and green space. Table 3 and figure 3 are confusing, and the legend of figure S.2 is not clear. Please check the full text. Authors should divide the “5. Discussion and conclusions” section into two sections. The manuscript could benefit from a careful editorial review on the language.

Author Response

Comments and Suggestions for Authors
9. Using the Thinking Fadura project as a case study, this paper attempts to assess the societal
profitability of removing barriers to access to urban greenspaces. The framework of
Participatory Benefit-Cost Analysis (PBCA) was developed to analyze the benefits and costs of
greenspaces, and multiple scenarios were designed to analyze the sensitivity of the results.
This study is interesting and I think it can contribute in literature. There are some suggestions
for further improvement.
R: The authors are very grateful for the positive comment.
10. Authors should add a subsection to clarify the key data sources used in this paper. How were
the participatory evaluation of stakeholders and experts, visitors and the general public aided
the traditional CBA?
R: Before the data collection methods were described in the Supplementary Material. Now the data
collection methods have been described in detail in section 3.2 Participatory evaluation and data
collection methods. See below new section:
“3.2. Participatory evaluation and data collection methods
The participatory evaluation followed a bottom-up approach in a sequential evaluation including three
main focal points of the analysis: key stakeholders and experts, visitors and the general public. This
assessment may provide a better understanding of the needs of implementing Thinking Fadura and
their impacts. The participation of stakeholders and experts, visitors and the general public also helped
to design a more precise economic evaluation.
13
3.2.1. Key stakeholders and experts
The first step in the PBCA was the assessment with key stakeholders and experts. A workshop with 20
participants was held in May 2018, including members from public services of different areas such as
social welfare, equality, multiculturalism, development cooperation, environment, urban planning,
housing, civil protection, economic promotion and health and Fadura sporting area. In the
development of the workshop there were two people facilitating the session.
The workshop had four objectives: i) to provide a better understanding of the context of Thinking
Fadura, ii) to identify the elements and the population affected by opening the greenspaces, iii) to
identify new uses in the greenspaces and relate them with potential impacts and iv) to assess the
potential impacts as a result of increasing accessibility in the greenspaces (more details in the
description of the workshop in the Supplementary Material).
3.2.2. Visitors
Subsequently, an assessment focusing on users of the greenspaces of Fadura was carried out between
August 2018 and June 2019. This included counting the users, an in-situ survey to visitors and an
assessment of physical activity.
3.2.2.1. Counting the users
The counting of visitors was done through manual counters at different times of the day, days of the
week and months of the year. The counting of visitors was needed for quantifying the recreation
benefit by the travel cost method [31].
3.2.2.2. In-situ survey to visitors
The in-situ survey to visitors provided quantitative data on frequency of use and potential increase in
the use of the park, travel cost values, and perceptions about the benefits and costs of Thinking Fadura
(for more detail see the Supplementary Material). Through this survey we obtained both qualitative
and quantitative information that was used for the different analysis carried out in this study.
The target population was adults who transited through the green areas that were currently open to
the public. A total of 256 visitors over 16 years old of age were surveyed.
3.2.2.3. Physical activity assessment
The assessment of physical activity was done with the System for Observing Play and Active Recreation
in Communities (iSOPARC, [32]) tool (see the SOPARC assessment in the Supplementary Material).
iSOPARC is an established and validated observation tool to assess the use of public spaces in
community settings [32]. iSOPARC is used to record individual (gender, age, physical activity level, and
ethnicity) and contextual characteristics (in a given area) and primary activity for each observed
person. Through visual scans, iSOPARC determines the usage and physical activity of the greenspace
at a determined time.
In Thinking Fadura, the iSOPARC assessment was carried out in order to determine the number of
people (all ages) increasing physical exercise as a result of the restoration of the pathway and the
improved accessibility in Fadura.
The green areas of Fadura (around 4ha) were divided in 5 target areas (see Figure S.6 in the
Supplementary Material). The observations were done by two observers in August 2018. In total
sixteen site visits were done in each Target Area (5 TAs) at different time of the day. These times were
at 9.00am, 12.00pm, 5.00pm and 8.00pm. The site visits were done in twelve different days covering
14
both weekdays and weekends. Table S.2 in the Supplementary Material shows the temporal
distribution at different times. Every site visit covered the five target areas.
In the five target areas, the iSOPARC tool counted an average of 2.05 people per scan. For the physical
exercise benefit only those people who were doing walking or vigorous activities were considered.
Thus in the five target areas, the average of people increasing physical exercise for walking and
vigorous activities were 1.06 and 0.21, respectively (Table S.3). Hence, the sum of people increasing
physical exercise was 1.27 (people / iSOPARC scan).
3.2.3. General public
The final step in the PBCA was the assessment of opinions among the general public. This was done
through a face-to-face survey in the municipality of Getxo (see the description of the survey to the
general public in the Supplementary Material). The aim of this survey was to study the opinion of the
citizens of Getxo with regards to Thinking Fadura´s initiative of opening Fadura´s greenspaces to the
general public. The target population was people over 16 years old and registered as citizens of the
municipality of Getxo. To identify this population, two screening questions were included in the
introduction of the survey. In total, 250 surveys were conducted in 12 different areas of the
municipality.”
This paper showed only two separate parts. Some items of participatory evaluation were not involved
in CBA. Why? The basis for selecting the respondents of this study should be further explained.
R: The participatory evaluation description has been significantly improved (see new section 3.2.
above). We believe that now the selection of the respondents and the target population has been
described.
11. Namely, the representativeness of samples. Authors need to provide more explanations for
their designed items of participatory evaluation. And how these items aid the CBA.
R: This was explained before in the Supplementary Material. It has now described in section 3.2
Participatory evaluation and data collection methods (see response to comment #10).
12. In the “3.4. Sensitivity analysis” subsection, all these 54 scenarios were analyzed using the
same basic surveys. What does this effect the analysis results? In different situations, will the
perception of the respondents not change? Please explain it.
R: The sensitivity analysis was implemented to assess the robustness of the economic assessment. We
focused on those parameters with the highest uncertainty. In our CBA, we considered that the
parameters with the highest uncertainty were the following:
 Discount rate: Three different discount rates were evaluated (0%, 3.5% and 10%)
 Time horizon: Three time horizons were included (10, 20 and 30 years)
 Assumed values for items with high uncertainty: These parameters are reported in Table 2.
Three scenarios were included: -20%, 0% and +20% of the assumed value of the items.
 Changes in future use: Since the greenspaces were only partially opened at the time when the
survey was implemented it was considered that future use could increase over time.
Throughout the survey to the general public, the percentage of respondents that would
increase their frequency of use once the park is fully opened was estimated (24.8%). This
criterion includes two scenarios: considering that future use will be the same, and considering
that future use will increase by 24.8%.
15
The four criteria led to a total of 54 distinct scenarios. In these scenarios, the benefits, costs and net
present value were assessed for the sensitivity analysis. The sensitivity analysis allowed us to see
which variables were more affected by the sources of uncertainty. This is shown in Figure 3.
13. The terms in the paper (both text, tables and figures) should be consistent, e.g. greenspaces,
and green space.
R: The terms in the papers have been changed to be consistent in text, tables and figures.
14. Table 3 and figure 3 are confusing, and the legend of figure S.2 is not clear. Please check the
full text.
R: The legends of Table 3, Figure 3 and Figure S.2 have been changed to clarify their meaning.
15. Authors should divide the “5. Discussion and conclusions” section into two sections.
R: Discussions and Conclusions section has now been divided into two separate sections. See
comments #7 and #8.
16. The manuscript could benefit from a careful editorial review on the language.
R: The English grammar has been reviewed by an English native speaker.

Reviewer 4 Report

Comments and Suggestions for Authors

In general I found the manuscript to be well-conceived and written. The methods used seem appropriate and the conclusions aligned with the results.

One overarching thought I had when reviewing this manuscript is that while I would agree that there are quantifiable benefits (and disbenefits) associated with urban greenspace and enhanced access to that space, the use of terminology such as "profitability", "cost effective" and "cash flow", etc. can be misinterpreted. From my perspective, these are more financial terms that require an entity that is assuming the cost or revenue. That is, cost to whom? As an example, the city would likely pay the cost for restructuring roads. However, the city does not accrue the social recreation benefit. At least not directly. I would be more comfortable with terminology that sticks to the analysis being an economic cost/benefit analysis with direct and indirect economic costs and benefits.

One possible disbenefit that is not mentioned is potential loss of revenue from the annual membership fee if members choose not to renew due to the park and facilities being more widely open. However, if membership stays the same and new revenue is gained by the introduction of one-day passes, that would be a positive economic benefit.

Author Response

Comments and Suggestions for Authors
17. In general I found the manuscript to be well-conceived and written. The methods used seem
appropriate and the conclusions aligned with the results.
R: The authors are thankful for the positive and valuable comments.
18. One overarching thought I had when reviewing this manuscript is that while I would agree
that there are quantifiable benefits (and disbenefits) associated with urban greenspace and
enhanced access to that space, the use of terminology such as "profitability", "cost effective"
and "cash flow", etc. can be misinterpreted. From my perspective, these are more financial
terms that require an entity that is assuming the cost or revenue. That is, cost to whom? As
an example, the city would likely pay the cost for restructuring roads. However, the city does
not accrue the social recreation benefit. At least not directly. I would be more comfortable
with terminology that sticks to the analysis being an economic cost/benefit analysis with
direct and indirect economic costs and benefits.
R: We agree with the reviewer that we have carried out an economic cost-benefit analysis as we are
measuring all benefits against the costs (market + non-market) in monetary units (Hutton and
Rehfuess, 2006). The Net Present Value (NPV) is used to express all flows of benefits and costs over
time on a common basis by taking into consideration the time when they are incurred (see Eq. 1 in the
manuscript). So, what we now conclude is whether this investment’s benefits outweigh its costs, and
by how much, as well as measuring the net effect of the Thinking Fadura intervention under different
secnarios.
17
Thus, the terminology has been amended along the manuscript accordingly, including the title of the
paper. We have deleted the words “The economic profitability” from the title and it has been updated
(now the title: “Providing access to urban greenspaces: A participatory benefit-cost analysis in Spain”).
We have avoided the use of terms such as profitability (and used instead NPV or when corresponding
say economic feasibility) or cost-effective (and instead say whether economic benefits outweigh the
costs).
Reference:
Hutton, H., Rehfuess, E., 2006. Guidelines for conducting cost-benefit analysis of household energy
and health interventions. WHO Library Cataloguing-in-Publication Data. World Health Organization.
19. One possible disbenefit that is not mentioned is potential loss of revenue from the annual
membership fee if members choose not to renew due to the park and facilities being more
widely open. However, if membership stays the same and new revenue is gained by the
introduction of one-day passes, that would be a positive economic benefit.
R: The authors appreciate this comment. There was no significant change in the revenue from the
annual membership fee as a result of the park opening. Hence neither disbenefits nor benefits could
be considered in the CBA. In any case, the following text has been inserted in the manuscript to clarify
this:
“Based on the membership data from the Fadura´s Municipal Sports Center, there was no significant
change in the revenue from the annual membership fee as a result of the park opening. Thus neither
disbenefits nor benefits associated to changes in the revenue from the annual membership fee were
considered in the CBA.”

Round 2

Reviewer 1 Report

The authors fulfilled all the reviewers' requests.

The manuscript is now much clearer and more complete, thanks also to the integration of many aspects previously contained only in the additional material.

Good job!

Reviewer 2 Report

All my comments have been revised, and this paper can be accepted in present form.

Reviewer 3 Report

The modifications made to the manuscript appear to be sound and increase the overall merit. Good work!